# Hydrogen Sulphide Treatment Prevents Renal Ischemia-Reperfusion Injury by Inhibiting the Expression of ICAM-1 and NF-kB Concentration in Normotensive and Hypertensive Rats

**DOI:** 10.3390/biom11101549

**Published:** 2021-10-19

**Authors:** Syed F. Hashmi, Hassaan Anwer Rathore, Munavvar A. Sattar, Edward J. Johns, Chee-Yuen Gan, Tan Yong Chia, Ashfaq Ahmad

**Affiliations:** 1School of Pharmaceutical Sciences, Universiti Sains Malaysia, Penang 11800, Malaysia; fayazhashmi84@gmail.com (S.F.H.); hrathore@qu.edu.qa (H.A.R.); munavvar@usm.my (M.A.S.); 2Department of Physiology, University College Cork, T12 K8AF Cork, Ireland; ej.johns@ucc.ie; 3Analytical Biochemistry Research Centre (ABrC), Universiti Sains Malaysia (USM), Lebuh Bukit Jambul, Penang 11900, Malaysia; cygan@usm.my; 4Department of Pharmacy Practice, College of Pharmacy, University of Hafr Al-Batin, Hafr Al-Batin 31991, Saudi Arabia

**Keywords:** hydrogen sulphide, ischemia-reperfusion injury, L-nitro-arginine-methyl-ester, ICAM-1, NF-kB, PAG

## Abstract

Our main objective was to investigate the effect of chronic administration of hydrogen sulphide donor (sodium hydrosulphide) on the expression of intercellular adhesion molecule-1 (ICAM-1) and concentration of nuclear factor-kappa B (NF-kB) in a renal ischemia-reperfusion injury (IRI) model of WKY and L-nitro-arginine-methyl-ester (L-NAME)-induced hypertensive rats. Sodium hydrosulphide (NaHS) was administered intraperitoneally (i.p.) for 35 days while cystathionine gamma lyase (CSE) inhibitor dL-propargylglycine (PAG) was administered at a single dose of 50 mg/kg. Animals were anesthetised using sodium pentobarbitone (60 mg/kg) and then prepared to induce renal ischemia by clamping the left renal artery for 30 min followed by 3 h of reperfusion. Pre-treatment with NaHS improved the renal functional parameters in both WKY and L-NAME-induced hypertensive rats along with reduction of blood pressure in hypertensive groups. Oxidative stress markers like malondialdehyde (MDA), total superoxide dismutase (T-SOD) and glutathione (GSH) were also improved by NaHS treatment following renal IRI. Levels of ICAM-1 and NF-kB concentration were reduced by chronic treatment with NaHS and increased by PAG administration after renal IRI in plasma and kidney. Treatment with NaHS improved tubular morphology and glomerulus hypertrophy. Pre-treatment with NaHS reduced the degree of renal IRI by potentiating its antioxidant and anti-inflammatory mechanism, as evidenced by decreased NF-kB concentration and downregulation of ICAM-1 expression.

## 1. Introduction

Ischemia-reperfusion injury (IRI) is a common cause of functional impairment of reperfused organs associated with increased production of reactive oxygen species (ROS). ROS during IRI induce initial inflammatory responses, which in turn results in the adhesion of leukocytes [1]. Leukocytes’ adhesion, rolling and migration to the ischemic tissues, followed by reperfusion, is initiated by initial inflammatory responses [2]. The expression of ICAM-1 on the surface of leukocytes [3] and endothelium [4] plays an important role in potentiating the initiation of inflammatory responses [5], which are involved in the pathogenesis of post-ischemic organ damage. Transcription of ICAM-1 is dependent upon nuclear factor-kappa B (NF-kB) activation [6]. In non-stimulated cells, NF-kB lies in the cytoplasm in inactive form, along with inhibitory kappa B (IkB) inhibitors. Various stimuli cause the degradation of IkB. As a result of this degradation, NF-kB enters the nucleus where it induces the synthesis of its specific mRNA by binding to DNA. Expressions of different specific genes are controlled by the transcription factor NF-kB and, once this transcription factor is activated by any stimuli, it can enhance the severity of IRI [7]. ROS production causes the activation of NF-kB [8,9]. Once NF-kB is activated, it induces ICAM-1 expression [10]. However, ROS and inflammatory responses play key roles in the pathogenesis of IRI through NF-kB and the ICAM-1 pathway.

The failure of many renal transplantation procedures results from the consequences of IRI. Various treatment approaches have been investigated by targeting oxidative stress and inflammation in order to minimize the extent of IRI in laboratory animals. Treatments, including antioxidant therapies and ROS scavengers, like naringin [11] and catechin [12], have been reported to improve renal IRI. Keeping in view the importance of antioxidant and anti-inflammatory mechanisms in IRI, the present study chose exogenous hydrogen sulphide (H_2_S) treatment because of its antioxidant [13,14] and anti-inflammatory potential [15,16] and its hypo-metabolism-induced hypoxic environment [17].

H_2_S is a physiologically active gasotransmitter. Endogenously, two enzymes, named cystathionine beta synthase (CBS) and cystathionine gamma lyase (CSE), are involved in the production of H_2_S [18]. However, a third enzyme named 3-mercaptopyruvate sulphur transferase (3-MST) has also recently been reported to participate in the production of H_2_S [19]. CSE is mainly involved in the production of H_2_S in kidneys and CVS while CBS controls H_2_S production in CNS [20]. Approximately, 90% of H_2_S production in the brain is carried out by 3-MST [21]. H_2_S is a vasorelaxant [19,22,23,24], anti-hypertensive [25,26,27], antioxidant [13,14,28,29] and anti-inflammatory agent [15,30]. Data also indicate that H_2_S may produce its therapeutic role as anti-inflammatory and antioxidant pathway in the kidney, which is dependent on the carbon monoxide pathway (CO) [31].

Acute administration of NaHS, either topically on to the kidneys [32,33] or through intravenous flushing in to the kidneys [34,35], has been studied in a renal IRI model in normotensive rats. However, to the best of our knowledge, the effects of chronic treatment with NaHS in a renal IRI model, both for normal and diseased models, are as-yet unknown. Thus, in the present study we investigated the effects of pre-treatment with NaHS for 35 days on renal function in both normotensive as well as L-NAME-induced hypertensive rats following IRI. The present study included an L-NAME-induced hypertensive model, as this model is characterized by increased oxidative stress [36,37] as well as decreased endogenous concentration of H_2_S [27]. Therefore, the kidneys of hypertensive rats would be more capable of rendering the extent of IRI compared to normotensive rats. We hypothesized that chronic administration of NaHS would improve the physiological concentration of H_2_S, which would induce its antioxidant potential by increasing antioxidant markers like T-SOD and GSH and decreasing pro-oxidant markers like MDA in the plasma of normotensive and hypertensive rats. This elevated antioxidant mechanism and inhibition of NF-kB activation would downregulate ICAM-1 expression through its anti-inflammatory potential. We hypothesized that enhancement of antioxidant and anti-inflammatory mechanisms by H_2_S would reduce the degree of IRI in normotensive and hypertensive rats.

## 2. Materials and Methods

### 2.1. Animals

Sixty-four male *Wistar Kyoto* rats (WKY) weighing 220 ± 20 g were received from the Animal Research and Service Centre (ARASC) at Universiti Sains Malaysia (USM) and brought to the transit room of the School of Pharmaceutical Sciences, USM, where they were acclimatized for 5 days. All the animals were housed in standard cages with free access to chow food (protein 22%; fat 3%; fibre: 3%; ash: 8%; calcium: 1%; phosphorus: 0.8–1.2%; sodium: 0.18–0.24%; potassium: 1.0–1.1%; and moisture: 13%) and water. All the experimental procedures were approved by the Universiti Sains Malaysia Animal Ethics committee with approval letter reference no. USM/Animal Ethics Approval/2015/(95) (649).

### 2.2. Induction of Hypertension

L-NAME was dissolved in drinking water at a dose of 40 mg/kg/day to induce hypertension [38,39]. The stock solution of L-NAME was prepared on a daily basis. L-NAME was administered for 28 days, started on day 8 until the end of the study protocol. The dose of L-NAME was adjusted according to body weight of the rats three times per week.

### 2.3. Exogenous Administration of NaHS

NaHS was administered i.p. at a dose of 56 µmol/kg daily for a period of 35 days [40,41]. Stock solution of NaHS was prepared daily by dissolving NaHS in normal saline (0.9% *w*/*v*). The dose of NaHS was adjusted according to body weight of the rats three times per week.

### 2.4. Administration of PAG

PAG was administered i.p. at a single dose of 50 mg/kg on day 36 (acute experiment day) [32]. PAG solution was prepared daily by dissolving PAG in normal saline (0.9% *w*/*v*).

### 2.5. Drugs and Chemicals

NaHS was purchased from the USA (Sigma-Aldrich) and L-NAME (Sigma-Aldrich, Schaffhausen, China), PAG (Sigma-Aldrich, Schaffhausen, Switzerland) and pentobarbital sodium (Dorminal 20%) from Holland (Alfasan, Woerden, The Netherlands).

### 2.6. Experimental Protocol

Animals were divided into two groups, namely WKY and L-NAME-induced hypertension. Each group was subdivided into four groups (*n* = 8, each group) consisting of (i) rats that received saline (WKY-SHAM and L-NAME-SHAM), (ii) rats that received saline (WKY-CONTROL and L-NAME-CONTROL), (iii) rats that received H_2_S treatment (WKY+NaHS and L-NAME+NaHS) and (iv) rats that received PAG treatment (WKY+PAG and L-NAME+PAG). All groups of rat underwent the same surgical procedure to induce IRI. The sham-operated group underwent the same surgical procedure, except that the IRI was not induced.

### 2.7. Collection of Metabolic Data

The animals were kept individually in metabolic cages (Nalgene^®^, Thermo Scientific, Beaumont, PA, USA) and body weight, 24 h urine output and water intake were measured (data not shown) on days 0, 21 and 35. In order to remove impurities, the urine samples were centrifuged at 10,000× *g* rpm (Hettich EBA 8S, Zentrifugen, Hettich Instruments, Illinois, Chicago, USA) for 3 min. Blood samples were also collected (300–500 µL) on days 0, 21 and 35 from the lateral tail vein and centrifuged at 10,000× *g* rpm for 10 min to separate clear plasma [42]. Both urine and plasma samples were stored at −30 °C for further biochemical estimation of creatinine, potassium and sodium on days 0, 21 and 35. Finally, on day 36 the acute experiments were performed to induce IRI.

### 2.8. Non-Invasive Blood Pressure (NIBP) Measurment

Non-invasive blood pressure (NIBP) was measured using the tail-cuff method (CODA^TM^, Kent Scientific Corporation, Torrington, CT, USA) in conscious animals on days 0, 21 and 35. Thereafter, invasive blood pressure was measured directly in anaesthetized animals on day 36 during the acute experimental procedure.

### 2.9. Acute Experiment

The experimental procedure for the acute surgical experiment was based on previously reported studies [42,43]. The rats were fasted overnight for 8–10 h and then anesthetized using pentobarbital sodium (60 mg/kg, i.p.). A tracheotomy was performed by inserting a tracheal tube PP 240 (Portex Ltd. Kent, London, UK) into the trachea to ensure clear airway passage. Thereafter, the left jugular vein was cannulated in order to administer a maintenance dose of anaesthesia at a dose of 12.5 mg/kg as required. The right carotid artery was also cannulated, and the cannula was connected to the pressure transducer system (P23 ID Gould, Statham Instruments, London, UK), which was linked to the data acquisition system (PowerLab^®^, ADInstruments, Sydney, Australia) via a Quad amp (ADInstruments, New South Wales, Australia) for continuous measurement of the mean arterial pressure (MAP), systolic blood pressure (SBP) and heart rate (HR) using Chart Pro (V.5.5) software on a Hewlett Packard Centrino Core2 Duo computer operating Windows XP. Afterwards, an abdominal midline incision was made to expose the left kidney and left iliac artery. The iliac artery was cannulated to measure iliac blood pressure (data not shown). The iliac cannula was also connected to a pressure transducer (P23 ID Gould, Statham Instruments, Nottingham, UK) coupled to a computerized data acquisition system (PowerLab, AD Instruments, Sydney, NSW, Australia). The iliac cannula was further connected to an infusion pump for the continuous infusion of normal saline (0.9% NaCl) at a rate of 3 mL/h. Finally, RCBP was measured using a laser Doppler flow probe (OxyFlow, ADInstruments, New South Wales, Australia), which was positioned on the dorsal surface of the posterior end of the exposed left kidney. The probe was connected to a laser Doppler flow meter (ADInstruments), which was directly linked to the data acquisition system (PowerLab, ADInstruments). The urinary bladder was catheterized to collect urine samples during the surgical procedure. On completion of surgery, the rat was given a period of 1 h for stabilization.

### 2.10. Collection of Blood and Urine Samples

After stabilization, baseline values of SBP, MAP, HR and RCBP were recorded. Blood (1.5 mL) was collected from the carotid artery cannula, centrifuged to obtain plasma and finally stored at −80 °C for future biochemical tests. In order to restore the withdrawn blood, the centrifuged blood without plasma was resuspended with 0.9% normal saline and infused slowly through a jugular vein cannula. A recovery period of 30 min was provided for stabilization in order for variables to return to baseline levels. Once the rat was stabilized, the left renal artery was clamped for 30 min with a non-traumatic arterial clamp (Tenko, Germany) in order to induce renal ischemia according to a previous reported procedure [44]. A 30 min ischemia was followed by a reperfusion phase of 3 h. At the end of the reperfusion phase, approximately 3–5 mL of blood was withdrawn from the carotid artery and centrifuged to separate plasma. Urine samples (2–3 mL) were also collected before and after renal ischemia. Both plasma and urine samples were stored at −80 °C for future biochemical tests. The animal was then euthanized with an overdose of pentobarbital sodium (200 mg/kg). The left kidney was then removed and weighed in order to calculate the kidney index according to an already reported method [42].

(1)
$$\mathrm{Kidney}\;\mathrm{Index}=\frac{\mathrm{Kidney}\;\mathrm{weight}}{\mathrm{Body}\;\mathrm{weight}}\times 100$$


### 2.11. Measurement of Renal Functional Parameters

Concentration of creatinine was measured in both plasma and urine using the colorimetric method according to an already published procedure [45]. Readings were taken at 520 nm using a 96-well microplate reader (Epoch^TM^ Microplate Spectrophotometer, BioTek Instruments, Einooski, Vermont, USA). Creatinine clearance was then calculated from the readings of urinary and plasma creatinine. Sodium and potassium in plasma and urine were measured using a flame photometer (Jenway Limited, Felsted, UK). These readings were then used to calculate the fractional excretion of sodium (FE_Na_) and fractional excretion of potassium (FE_K_).

### 2.12. Measurement of H_2_S Concentration

H_2_S concentration in plasma was measured spectrophotometrically [40,46] on days 0, 21 and 36. The procedure involved the addition of 300 µL zinc acetate (1% *w*/*v*) into centrifuge tubes, and then 100× *g* µL of plasma samples and 50 µL of distilled water were added into the tubes. The purpose of the zinc acetate in the procedure was to trap H_2_S. Then, 200 µL of N,N-dimethyl-P-phenylenediamine sulphate (20 mM in 7.2 M HCl) was added after 5 min in order to stop the reaction. This step was immediately followed by the addition of 200 µL ferric chloride (30 mM in 1.2 M HCl). All the samples were then kept in the dark for 20 min. Then, 150 µL of trichloro acetic acid (10% *w*/*v*) was added to all samples in order to precipitate out the proteins. Finally, samples were then centrifuged and readings of supernatant were taken at 670× *g* nm. The concentration of H_2_S in the plasma was calculated by using the standard curve for H_2_S.

### 2.13. Measurement of Oxidative Stress Markers

Oxidative stress markers in plasma, such as malondialdehyde (MDA), total superoxide dismutase (T-SOD) and glutathione (GSH), were measured on day 36 (acute experiment day) at the pre-ischemia and reperfusion phases using MDA, T-SOD and GSH commercial kits (Institute of Biological Engineering of Nanjing Jianchen, Nanjing, China). All the procedures for these kits were followed according to the instructions given in their respective manuals.

### 2.14. ICAM-1 and NF-kB Measurement in the Plasma and Kidney Tissues of WKY and L-NAME-Treated Groups

ICAM-1 and NF-kB concentrations were measured in plasma and kidney tissues of all animals of the WKY and L-NAME groups at the pre-ischemia and reperfusion phases on day 36. Kidney levels of ICAM-1 and NF-kB in the pre-ischemia phase were measured in different sets of animals and the experiment was terminated after taking kidney samples. For quantitative measurement of ICAM-1 levels, a Quantikine ELISA Rats ICAM-1/CD54 ELISA kit (R&D Systems, Inc., USA) was used. The principle of this assay kit follows that of the quantitative sandwich enzyme immunoassay technique. Similarly, for quantitative measurement of rat NF-kB concentration in the plasma and kidney tissue, a Rat Nuclear Factor-Kappa B (NF-kB) ELISA kit (CUSABIO Biotech Co., Ltd.) was used. This assay kit also followed the quantitative sandwich enzyme immunoassay technique. All the procedures for both the Rat sICAM-1/CD 54 immunoassay and Rat Nuclear Factor-Kappa B (NF-kB) immunoassay were followed according to the instructions given in the manuals. The optical density of each sample for both ICAM-1 and NF-kB was measured spectrophotometrically at 450 nm. Then, the concentrations of ICAM-1 and NF-kB in the plasma and kidney tissue samples were calculated against their respective standard curves.

### 2.15. Histopathology of Rat Kidney Tissues of WKY and L-NAME Groups

Histopathology of kidney tissue followed a previously reported procedure [41]. Briefly, kidneys of all groups of WKY and L-NAME rats undergoing ischemia-reperfusion injury were extracted at the end of the experiments and placed in the 10% formalin. Kidneys were subjected to subsequent events of embedding, trimming, sectioning and staining with haematoxylin and eosin.

### 2.16. Statistical Analysis

All the data (mean ± SEM) were analysed statistically using the GraphPad statistical software package (GraphPad Prism Statistics for Windows; version 5.0, San Diego, CA, USA). Data were analysed and compared using repeated measures one-way ANOVA followed by Bonferroni’s post hoc test. For all comparisons, differences between the means were considered significant when *p* < 0.05.

## 3. Results

### 3.1. Systemic Hemodynamic Parameters

SBP, MAP and HR were measured on days 0, 21 and 35 with a non-invasive method in both WKY and L-NAME-induced hypertensive rats (Table 1). In the WKY-CONTROL groups no significant changes in SBP, MAP or HR were observed on days 21 and 35 compared to their respective groups on day 0. On the other hand, SBP and MAP were significantly increased (all *p* < 0.05) in L-NAME-induced hypertensive groups on days 21 and 35 when compared to their respective groups on days 0. Similarly, HR in L-NAME-induced hypertensive groups was significantly increased (all *p* < 0.05) on day 35 compared to both days 0 and 21. In the L-NAME+NaHS group, HR was significantly increased (*p* > 0.05) on day 35 compared to days 0 and 21. Moreover, SBP and MAP were significantly increased (all *p* < 0.05) in L-NAME-induced hypertensive rats on days 21 and 35 compared to the WKY-CONTROL group on days 21 and 35.

### 3.2. Renal Functional Parameters

FE_Na_, FE_K_ and creatinine clearance (Cr.Cl) were measured in both WKY and L-NAME-induced hypertensive groups on days 0, 21 and 35 (Table 1). In WKY groups no significant change (all *p* > 0.05) in FE_Na_, FE_K_ and Cr.Cl was observed on day 21 when compared to their respective groups on day 0, nor on day 35 when compared to their respective groups on both days 0 and 21. On the other hand, FE_Na_ and FE_K_ were significantly increased (all *p* < 0.05) in L-NAME-induced hypertensive groups on day 21 compared to their respective groups on day 0 and on day 35 when compared to their respective groups on both days 0 and 21, except the L-NAME+NaHS group, in which both FE_Na_ and FE_K_ were significantly increased (*p* < 0.05) on day 35 compared to day 0 but not significantly increased when compared to day 21. Pre-treatment with NaHS significantly reduced (*p* < 0.05) both FE_Na_ and FE_K_ in the L-NAME+NaHS group when compared to its L-NAME control group on day 35. Cr.Cl in the L-NAME-induced hypertensive groups was significantly reduced (all *p* < 0.05) on day 21 as well as on day 35 when compared to their respective groups on day 0, except in the L-NAME+NaHS group in which no significant changes (*p* > 0.05) in Cr.Cl were observed on days 21 or 35 compared to day 0. Moreover, FE_Na_ and FE_K_ were significantly increased (*p* < 0.05), while Cr.Cl was significantly reduced (*p* < 0.05), in L-NAME-induced hypertensive rats on day 35 compared to WKY rats on the same day.

### 3.3. H_2_S Concentration in Plasma

The plasma concentration of H_2_S in both WKY and L-NAME-induced hypertensive groups was measured on days 0, 21 and day 36 (Figure 1). In WKY groups, no significant change (all *p* > 0.05) in plasma H_2_S concentration was observed on day 21 when compared to their respective groups on day 0 or on day 36 when compared to their respective groups on both days 0 and 21, except the WKY+NaHS group, which was pre-treated with NaHS and showed a significant increase (*p* < 0.05) in plasma concentration of H_2_S on day 36 compared to its respective group on day 0 and also to the WKY-CONTROL group on the same day. On the other hand, administration of PAG in the WKY+PAG group showed a significant decrease (*p* < 0.05) in plasma concentration of H_2_S on day 36 when compared to its respective group on day 0 and also to the WKY-CONTROL and WKY+NaHS groups on the same day. In the L-NAME-induced hypertensive groups, concentration of H_2_S in the plasma was reduced significantly (all *p* < 0.05) on day 21 when compared to their respective groups on day 0 and on day 36 when compared to their respective groups on both days 0 and 21, except the L-NAME+NaHS group in which there was no significant change (*p* > 0.05) in plasma H_2_S concentration on day 36 compared to days 0 and 21, but the H_2_S concentration in the L-NAME+NaHS group was found to be significantly increased (*p* < 0.05) on day 36 when compared to the L-NAME-CONTROL and L-NAME+PAG groups on the same day. The concentration of H_2_S in plasma was significantly lower (*p* < 0.05) in the L-NAME-induced hypertensive rats on day 36 compared to WKY rats on the same day.

### 3.4. Systemic Hemodynamics on Acute Experiment Day

The SBP, MAP and HR were also measured on acute experiment day using an invasive method (Table 2). All groups of WKY rats showed no significant differences (all *p* < 0.05) in SBP, MAP and HR. However, the same haemodynamic parameters were increased significantly (all *p* < 0.05) in L-NAME-induced hypertensive rats compared to WKY rats. The L-NAME group, when pre-treated with NaHS (L-NAME+NaHS), showed significantly reduced SBP, MAP and HR (all *p* < 0.05) compared to the L-NAME-CONTROL and L-NAME+PAG groups.

### 3.5. Body Weight, Kidney Weight and Kidney Index

The body weight (BW), kidney weight (KW) and kidney index (KI) were also measured on acute experiment day (Table 2). Both the KW and KI were significantly higher (both *p* < 0.05) in the CONTROL and PAG-administered groups in both WKY and L-NAME-induced hypertensive groups. It was observed that pre-treatment with NaHS did not exert any significant effect on KW and KI except in the WKY+NaHS group, in which KW was significantly reduced (*p* < 0.05) when compared to the WKY-CONTROL and WKY+PAG groups. KW was significantly high (*p* < 0.05) in the L-NAME+PAG group when compared to the L-NAME-CONTROL and L-NAME+NaHS groups. Similarly, KI was also significantly high (*p* < 0.05) in the L-NAME+PAG group when compared to the L-NAME+NaHS group. It was also observed that BW and KI were significantly high (*p* < 0.05) in L-NAME-induced hypertensive rats when compared to WKY rats.

### 3.6. RCBP in the Pre-Ischemia and Reperfusion Phases

Renal cortical blood perfusion (RCBP) was measured at the pre-ischemia and reperfusion phases (Figure 2). A significant decline (all *p* < 0.05) in RCBP was observed at the reperfusion phase in both the WKY and L-NAME-induced hypertensive groups when compared to their respective groups at the pre-ischemia phase, except in the SHAM groups of both WKY and L-NAME-induced hypertensive rats, in which no ischemia was induced. No significant effect (*p*> 0.05) of NaHS was observed on RCBP in the WKY+NaHS group when compared to the WKY-CONTROL and WKY+PAG groups. However, on the other hand, a significant increase (*p* < 0.05) in RCBP was found in the L-NAME+NaHS group when compared to the WKY-CONTROL and WKY+PAG groups. It was noticed that RCBP was significantly lower (*p* < 0.05) in L-NAME-induced hypertensive rats compared to WKY rats.

### 3.7. Renal Functional Parameters at the Pre-Ischemia and Reperfusion Phases

Renal functional parameters like FE_Na_, FE_K_ and plasma creatinine were also measured at the pre-ischemia and reperfusion phases (Figure 3). A significant increase (all *p* < 0.05) in FE_Na_ was observed at the reperfusion phase in the L-NAME-CONTROL and also in the PAG-administered groups in both the WKY and L-NAME-induced hypertensive groups (WKY+PAG and L-NAME+PAG) when they were compared to their respective groups at the pre-ischemia phase. However, the rest of the groups did not show any significant changes (all *p* > 0.05) in FE_Na_ at the reperfusion phase when they were compared to their respective groups at the pre-ischemia phase. At the reperfusion phase in the WKY groups, FE_Na_ was significantly reduced (*p* < 0.05) in the WKY+NaHS group when compared to the WKY+PAG group at the same phase. Similarly, in L-NAME-induced hypertensive groups, FE_Na_ was also found to be significantly reduced (*p* < 0.05) in the L-NAME+NaHS group when compared to the L-NAME-CONTROL and WKY+PAG groups at the reperfusion phase. No significant changes (all *p* > 0.05) in FE_K_ or plasma creatinine were observed in either the WKY or the L-NAME-induced hypertensive rats at the reperfusion phase when they were compared to their respective groups at the pre-ischemia phase. However, FE_K_ and plasma creatinine were found to be significantly reduced (*p* < 0.05) in the L-NAME+NaHS group when compared to the L-NAME-CONTROL and L-NAME+PAG groups. It was also observed that FE_Na_, FE_K_ and plasma creatinine were significantly high (*p* < 0.05) in L-NAME-induced hypertensive rats when compared to WKY rats.

### 3.8. Oxidative Stress Markers at the Pre-Ischemia and Reperfusion Phases

Oxidative stress markers such as MDA, T-SOD and GSH were also measured at the pre-ischemia and reperfusion phases (Figure 4). No significant changes (*p* > 0.05) were observed in oxidative stress markers in the SHAM groups in which no ischemia was induced. A significant increase (*p* < 0.05) in MDA concentration was observed only in the WKY-CONTROL and L-NAME-CONTROL groups at the reperfusion phase when compared to their respective groups at the pre-ischemia phase. Pre-treatment with NaHS and administration of PAG did not exert any significant effect on the MDA concentration at the reperfusion phase when compared to their respective groups at the pre-ischemia phase. A significant decrease (all *p* < 0.05) in T-SOD level was observed at the reperfusion phase in CONTROL and PAG-administered groups both in WKY and L-NAME-induced hypertensive rats when compared to their respective groups at the pre-ischemia phase. However, in groups pre-treated with NaHS, the T-SOD level was found to be significantly high (*p* < 0.05) at the reperfusion phase in both WKY and L-NAME-induced hypertensive groups (WKY+NaHS and L-NAME+NaHS) when compared to their CONTROL groups (WKY-CONTROL and L-NAME-CONTROL) and to PAG-administered groups (WKY+PAG and L-NAME+PAG) at the same phase. No significant changes (all *p* > 0.05) in GSH concentration were observed at the reperfusion phase in any WKY and L-NAME-induced hypertensive groups when they were compared to their respective groups at the pre-ischemia phase. However, it was observed that GSH concentration at the reperfusion phase was significantly high (*p* < 0.05) in the WKY+NaHS group compared to the WKY+PAG group at the same phase. Similarly, GSH concentration was also significantly high (*p* < 0.05) at the reperfusion phase in the L-NAME+NaHS group when compared to the L-NAME-CONTROL and L-NAME+PAG groups. It was observed that MDA concentration was significantly higher (*p* < 0.05) and T-SOD and GSH levels were significantly lower (all *p* < 0.05) in L-NAME-induced hypertensive rats when compared to WKY rats.

### 3.9. ICAM-1 Expression and NF-kB Concentration at the Pre-Ischemia and Reperfusion Phases in Plasma and Kidney Tissues of Different Groups

Levels of ICAM-1 and NF-kB concentration were also measured at the pre-ischemia and reperfusion phases in both WKY and L-NAME-induced hypertensive groups, as shown in Figure 5A,B. No significant differences (*p* > 0.05) in either ICAM-1 levels or NF-kB concentration were noticed in the SHAM groups of either WKY or L-NAME-induced hypertensive groups in which no ischemia was induced. However, significant increases (all *p* < 0.05) in ICAM-1 levels and NF-kB concentration were observed in both WKY and L-NAME-induced hypertensive groups at the reperfusion phase when they were compared to their respective groups at the pre-ischemia phase, except the NaHS treatment groups (WKY+NaHS and L-NAME+NaHS), in which there was no significant increase (*p* > 0.05) in NF-kB concentration at the reperfusion phase compared to their respective groups at the pre-ischemia phase. Pre-treatment with NaHS significantly reduced (all *p* < 0.05) levels of ICAM-1 and concentrations of NF-kB in both WKY and L-NAME-induced hypertensive rats (WKY+NaHS and L-NAME+NaHS) at the reperfusion phase when they were compared to their respective CONTROL groups (WKY-CONROL and L-NAME-CONTROL) at the same phase. Similarly, administration of PAG caused significant increases (*p* < 0.25) in ICAM-1 expression in both WKY and L-NAME-induced hypertensive groups (WKY+PAG and L-NAME+PAG) at the reperfusion phase when they were compared to their respective control groups (WKY-CONTROL and L-NAME-CONTROL) and also to the NaHS-treated groups (WKY+NaHS and L-NAME+NaHS) at the same phase. Similarly, in PAG-administered groups, significant increases (*p* < 0.05) in NF-kB concentration were observed in both the WKY and L-NAME-induced groups (WKY+PAG and L-NAME+PAG) at the reperfusion phase compared to their respective NaHS-treated groups (WKY+NaHS and L-NAME+NaHS) at the same phase. It was noticed that the ICAM-1 levels and NF-kB concentration were significantly high (*p* < 0.05) in L-NAME-induced hypertensive rats when compared to WKY rats.

Levels of ICAM-1 and NF-kB concentration were also measured at the pre-ischemia and reperfusion phases in the kidney tissues of both WKY and L-NAME-induced hypertensive groups, as shown in Figure 6A,B. The trends for the levels of ICAM-1 and NF-kB concentrations in the kidney tissue were the same as those in the plasma samples of the different groups. Induction of reperfusion significantly raised (*p* < 0.05) the levels of ICAM-1 in the kidneys of control and PAG-treated animals in the WKY and L-NAME groups, while chronic treatment with H_2_S donation to animals in the WKY and L-NAME groups significantly reduced (*p* < 0.05) the ICAM-1 levels in kidney tissue at the reperfusion phase when compared to the control WKY and control L-NAME groups at the same phase, as shown in Figure 6A.

Induction of reperfusion significantly raised (*p* < 0.05) the levels of NF-ҝB in the kidneys of control and PAG-treated animals in the WKY and L-NAME groups, while chronic treatment with H_2_S donation to the animals in the WKY and L-NAME groups significantly reduced (*p* < 0.05) the NF-ҝB levels in the kidney tissue in the reperfusion phase compared to control WKY and control L-NAME groups at the same phase, as shown in Figure 6B.

### 3.10. Histopathology of Kidney Tissue for All Animals in the WKY and L-NAME Groups (Sham, Control, NAHS and PAG)

The renal tissue of control rats showed normal glomerulus architecture and distal and proximal convoluted tubules of the nephron, as shown in Figure 7A. The renal tissue of ischemia reperfusion (I/R) affected the rats’ kidneys and indicated the presence of swollen renal tubules, with epithelial denudation of the basement membrane and a few necrotic tubules, as highlighted by the white arrows in Figure 7B. Renal tissue of sham-control rats showed a normal glomerulus and tubular architecture, as shown in Figure 7C. The renal tissue of H_2_S-treated rats showed normal renal tissue architecture, as shown in Figure 7D. Normal tubular morphology, still with a few swollen tubules, was observed, but the structure of renal tissue was almost similar to that of sham-control rats (C), without any glomerular hypertrophy or necrotic tubules, as shown in Figure 7E.

## 4. Discussion

The present study investigated the effect of exogenous administration of H_2_S in the attenuation of the extent of IRI in normotensive and L-NAME-induced hypertensive rats. This study proposed that exogenous administration of H_2_S would attenuate the extent of IRI through its antioxidant mechanisms by enhancing antioxidant markers and through its anti-inflammatory mechanism by reducing I-CAM expression and NF-kB concentration in both models of rats. The present study resulted in two key findings: first, exogenous administration of H_2_S reduced the extent of IRI by improving antioxidant markers such as SOD and GSH and attenuating the MDA in normotensive and L-NAME-induced hypertensive rats. The second novel observation was that the exogenous administration of H_2_S attenuated the extent of IRI by reducing I-CAM expression and NF-kB concentration due to its anti-inflammatory mechanism in normotensive and L-NAME-induced hypertensive rats.

Inflammation has been reported to be involved in the pathogenesis of injuries induced by ischemia reperfusion, and the infiltration of neutrophils is a main contributor in this regard [47], further increasing the severity of IRI by producing more ROS [48]. Reperfusion of blood induces the production of ROS, which play a major role in the sequence of IRI [49,50,51,52] by inducing the expression of ICAM-1 via NF-kB activation [53]. To the best of our knowledge, there is no information available regarding the antioxidant and anti-inflammatory potential of chronic treatment with an H_2_S donor (NaHS) on ICAM-1 expression and NF-kB activation in normotensive and hypertensive models following renal IRI.

In the present study, L-NAME rats showed an increase in MAP that was comparable to results reported previously [54,55,56]. Increased MAP in L-NAME rats was assumed to be due, in part, to its eNOS- and nNOS-inhibiting properties, which can result in NO deficiency and stimulation of the sympathetic nervous system [57]. Moreover, activation of the renin angiotensin system (RAS) has also been documented in an L-NAME-induced hypertension model [58]. However, when L-NAME groups of rats were pre-treated with an H_2_S donor (NaHS), a drop in MAP was observed. This reduction in blood pressure may have been due to the vasodilating properties of H_2_S [25,59]. The anti-hypertensive property of H_2_S in L-NAME-induced hypertension is well-documented and in line with previously reported data [27]. Increased HR in L-NAME-induced hypertension was also observed, and it has been reported that elevated sympathetic activity and oxidative stress may be responsible for increased HR in L-NAME-induced hypertension [56]. Similarly, increases in FE_Na_ and FE_K_ were observed in L-NAME-induced hypertensive rats, which is coherent with previously reported data [60]. These increases may have been due to glomerular injury and impaired renal sodium and potassium handling from chronic inhibition of NOS by L-NAME treatment, as L-NAME has also previously been reported as a causative agent for glomerular injury [61]. Surprisingly, decreases in both FE_Na_ and FE_K_ were found in the present study when the L-NAME group was pre-treated with NaHS. Reduced tubular damage in the kidneys has been reported following H_2_S supplementation [62]. An important renal functional parameter, Cr.Cl, which is an indirect indicator of the glomerular filtration rate, was also ameliorated in all L-NAME-induced hypertensive groups, which is in line with an earlier study [63].

The physiological concentration of H_2_S in rat serum is 46 µM [59], while other studies have reported a range of 50–160 µM in humans, rats and bovines [64,65]. In the present study, pre-treatment with NaHS in the WKY group for 35 days increased the plasma concentration of H_2_S but, in contrast, when WKY rats were administered with PAG, a reduction in plasma H_2_S concentration was observed. This finding is well-supported by previous studies [66,67,68,69]. Furthermore, L-NAME-induced hypertensive groups showed reductions in plasma H_2_S concentrations, but this was surprisingly improved following NaHS treatment. L-NAME is reported to cause inhibition of CSE gene expression, which in return reduces the endogenous production of H_2_S [27].

To the best of our knowledge, there is no information available regarding RCBP data for renal IRI. RCBP was reduced in all groups of WKY and L-NAME-induced hypertensive rats at the reperfusion phase, except in sham groups in which no ischemia was induced. The present findings are in accordance with earlier studies that have reported that renal blood flow is reduced after renal IRI [34,44]. However, when the L-NAME group was pre-treated with NaHS, the RCBP was increased, which was possibly due to the vasodilatory effect of H_2_S [59]. It is also likely that there was H_2_S-mediated enhancement of pre-glomerular arterial vasodilation and renal blood flow [70]. Another reason may have been the upregulation of the CSE/H_2_S pathway in the kidney due to exogenous administration of H_2_S, as reported previously [71]. Increases in kidney weight and kidney index (KI) were observed in both WKY and L-NAME-induced hypertensive rats. The present findings are supported by earlier studies, which have reported that increases in kidney weight are due to reflection of oedema, obstruction of peritubular capillaries due to leukocyte plugging and neutrophil infiltration into the injury site after renal IRI [72,73,74]. However, groups that were pre-treated with NaHS did not show such increases in either kidney weight or KI after renal IRI.

During the acute experiment, following IRI, only an increase in FE_Na_ was observed without any change in FE_K_ or plasma creatinine in the L-NAME-induced hypertensive groups, while NaHS treatment in the L-NAME group improved the FE_Na_ after renal IRI. Elevated FE_Na_ is due to impaired sodium-handling capacity in renal tubules after renal IRI. This finding is in line with a previously reported study that found that renal tubular cells are particularly vulnerable to IRI, which causes higher levels of sodium excretion [34].

Oxidative stress is an important feature in IRI that is followed by free-radical production [75] and renal damage characterized by lipid peroxidation of polyunsaturated fatty acids [76]. Lipid peroxidation involves the production of the highly toxic reactive aldehydic metabolite, MDA [77]. In the present study, increased MDA concentrations were observed after IRI in untreated control groups of both WKY and L-NAME rats. This finding is consistent with previous published studies on the same IRI animal model [78,79,80,81,82,83]. Interestingly, increased levels of MDA were observed following NaHS treatment. Furthermore, reductions in both T-SOD and GSH levels were observed in both WKY and L-NAME-induced hypertensive groups, except the NaHS-treated groups, after renal IRI. Supplementation of H_2_S increased both T-SOD and GSH levels in both groups after renal IRI. The increased MDA and decreased T-SOD levels indicate the imbalance between the antioxidant and pro-oxidants induced by renal IRI. Pre-treatment with NaHS improved the levels of all oxidative stress markers, including MDA, T-SOD and GSH levels, likely due to the antioxidant and scavenging properties of H_2_S, which is known to scavenge superoxide anions [84] and peroxynitrite [85]. On the other hand, when both groups of WKY and L-NAME rats were administrated with PAG, they showed decreased levels of T-SOD and GSH after renal IRI. The possible reason for the reduction in these two antioxidant enzymes is that PAG reduces the endogenous concentration of H_2_S; thus, the kidney becomes more capable of rendering oxidative stress. In short, L-NAME-induced hypertensive rats showed high levels of oxidative stress following ischemia. This finding is well-documented and supported by previously reported data [36,37].

One important finding of the study related to exploring the levels of ICAM-1 and NF-kB following renal IRI in plasma and kidney tissue to evaluate the events happening in the plasma and kidney tissue. Massive accumulation and infiltration of leukocytes occurs in IRI [86,87]. The results of the present study showed increased ICAM-1 expression in both WKY and L-NAME-induced hypertensive rats after renal IRI, except in SHAM groups that were not subjected to renal ischemia. The present findings are consistent with earlier studies [32,44]. Increased concentrations of ICAM-1 in the plasma and kidney tissue indicate the involvement of inflammation in the present model of renal IRI. However, when the same group model was pre-treated with NaHS, it reduced ICAM-1 concentration in the kidney after renal IRI. The present findings are in line with earlier published studies, which have also reported that exogenous administration of H_2_S reduces ICAM-1 expression in renal IRI. [32,33,35]. However, administration of PAG increased ICAM-1 expression in both groups. To the best of our knowledge, there are no available data regarding the effect of PAG on ICAM-1 expression in a renal IRI model for normotensive as well as L-NAME-induced hypertensive rats. However, overexpression of ICAM-1 has been reported following PAG administration in hepatic [69] and myocardial IRI models [68]. The present findings regarding overexpression of ICAM-1 through decreased endogenous concentration of H_2_S by PAG administration strengthen the notion that decreased endogenous concentration of H_2_S renders the reperfused kidney more prone to the consequences of IRI and that supplementation of H_2_S reduces the severity of IRI due to its anti-inflammatory potential, as evidenced by decreased expression of ICAM-1 in the plasma and kidney.

Increases in NF-kB concentrations in plasma and kidneys were also observed in both WKY and L-NAME-induced hypertensive rats after renal IRI, except in SHAM groups in which no ischemia was induced. Increased NF-kB concentration has also been reported previously in renal IRI models [32,88,89,90,91,92,93]. Pre-treatment with NaHS reduced NF-kB concentration after IRI in normotensive and hypertensive rats both globally and locally in the kidney. Decreased NF-kB concentration has also been reported in a renal IRI model following topical administration of NaHS onto the kidneys [32]. However, NF-kB concentration was increased after administration of PAG. Since, ROS have been reported to induce NF-kB activation, it can therefore be concluded from the present findings that H_2_S scavenged ROS through its antioxidant potential [94]. The interesting and novel finding of the present study was the inhibition of NF-kB activation by NaHS treatment, which in response reduced the expression of ICAM-1. PAG administration also confirmed that a decreased concentration of H_2_S is endogenously responsible for the increased NF-kB concentration and overexpression of ICAM-1 in normotensive and hypertensive rats following renal IRI. Histopathological examination of kidney tissue treated with NaHS also indicated normal tubular morphology, still with a few swollen tubules, but the structure of the renal tissue was almost similar to that of sham-control rats, without any glomerular hypertrophy or necrotic tubules.

## 5. Conclusions

Taken together, the results of this study indicate that exogenous administration of H_2_S reduced the extent of IRI by improving antioxidant markers, such as T-SOD and GSH, and attenuating the MDA in normotensive and L-NAME-induced hypertensive rats. In addition to this, exogenous administration of H_2_S attenuated the extent of IRI by reducing I-CAM expression and NF-kB concentration both locally in the kidney and systemically in normotensive and L-NAME-induced hypertensive rats due to its anti-inflammatory mechanism. The outcome of the present study will help in providing renal protection following renal IRI and will reduce the chances of transplanted kidney injuries. Exogenous administration of H_2_S not only provides renal protection in IRI but also improves the excretory function and morphology of the kidney.

## Figures and Tables

**Figure 1 biomolecules-11-01549-f001:**
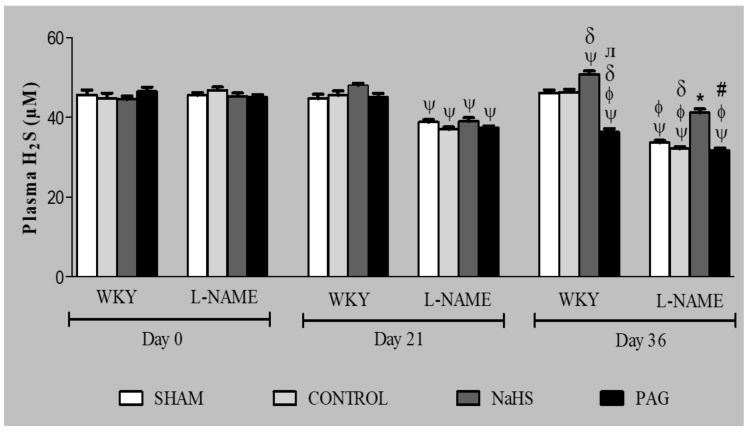
Plasma H_2_S concentrations on days 0, 21 and 36 (*n* = 6) in different experimental groups of WKY and L-NAME rats. Data were analysed using repeated measures ANOVA followed by Bonferroni’s post hoc test. ^Ψ^
*p* < 0.05 vs. day 0 of respective groups; ^φ^
*p* < 0.05 vs. day 21 of respective groups; ^δ^
*p* < 0.05 vs. WKY-CONTROL on day 36; ^ᴫ^
*p* < 0.05 vs. WKY+NaHS on day 36; * *p* < 0.05 vs. L-NAME-CONTROL on day 36; ^#^
*p* < 0.05 vs. L-NAME+NaHS on day 36. H_2_S = hydrogen sulphide, WKY = Wistar Kyoto rats, L-NAME = L-nitro-methyl-arginine-ester, PAG = dL-propargylglycine.

**Figure 2 biomolecules-11-01549-f002:**
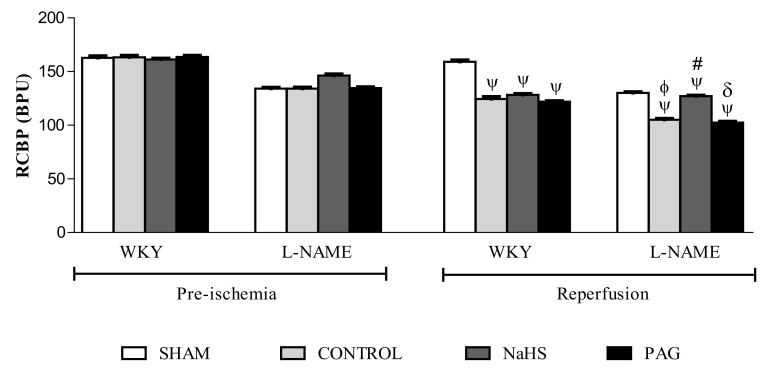
RCBP measurements in different experimental groups of WKY and L-NAME rats before and after ischemia (*n* = 6). Data were analysed using one-way ANOVA followed by Bonferroni’s post hoc test. ^Ψ^
*p* < 0.05 vs. pre-ischemic phase of respective groups; ^φ^
*p* < 0.05 vs. WKY-CONTROL at reperfusion phase; ^#^
*p* < 0.05 vs. L-NAME-CONTROL at reperfusion phase; ^δ^
*p* < 0.05 vs. L-NAME+NaHS at reperfusion phase. RCBP = renal cortical blood perfusion, WKY = Wistar Kyoto rats, PAG = dL-propargylglycine, L-NAME = L-nitro-methyl-arginine-ester.

**Figure 3 biomolecules-11-01549-f003:**
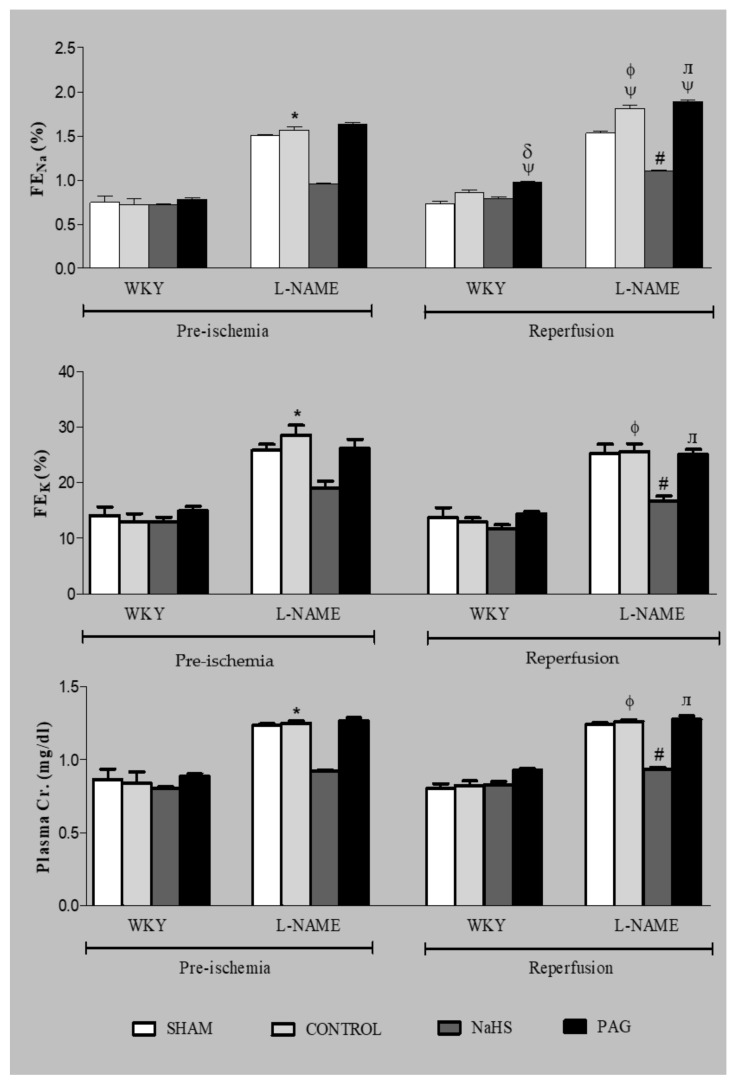
FE_Na_, FE_K_ and plasma creatinine measurements for different experimental groups of WKY and L-NAME rats before and after ischemia (*n* = 6). Data were analysed using one-way ANOVA followed by Bonferroni’s post hoc test. ^Ψ^
*p* < 0.05 vs. pre-ischemic phase of respective groups; * *p* < 0.05 vs. WKY-CONTROL at pre-ischemic phase; ^φ^
*p* < 0.05 vs. WKY-CONTROL at reperfusion phase; ^δ^
*p*< 0.05 vs. WKY+NaHS at reperfusion phase; ^#^
*p* < 0.05 vs. L-NAME-CONTROL at reperfusion phase; ^ᴫ^
*p* < 0.05 vs. L-NAME+NaHS at reperfusion phase. FE_Na_ = fractional excretion of sodium, FE_K_ = fractional excretion of potassium, Cr. = creatinine, PAG = dL-propargylglycine, WKY = *Wistar Kyoto* rats, L-NAME = L-nitro-methyl-arginine-ester.

**Figure 4 biomolecules-11-01549-f004:**
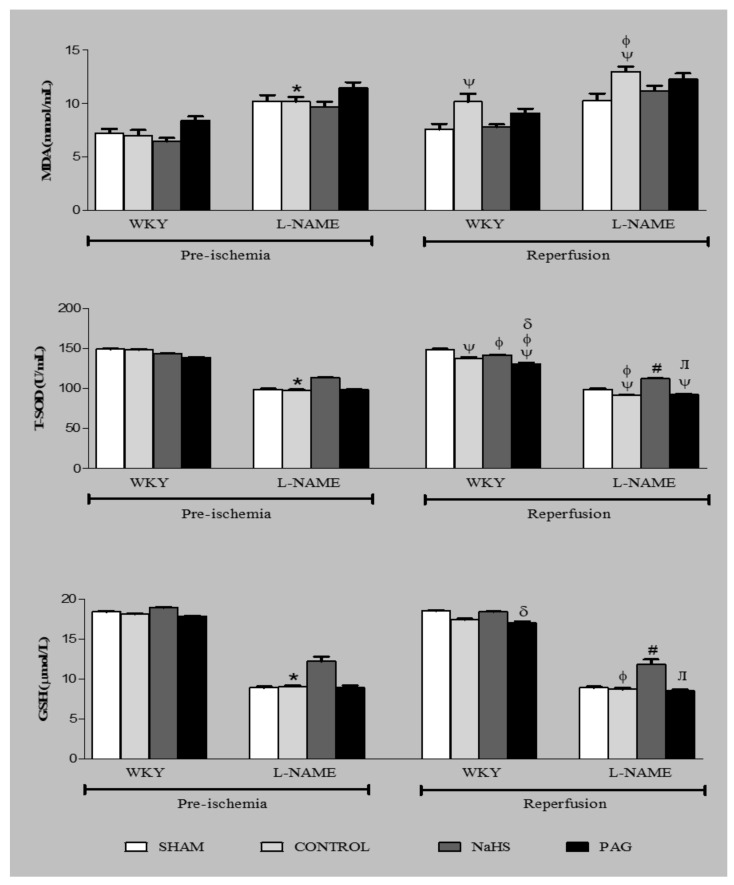
MDA, T-SOD and GSH concentrations in different experimental groups of WKY and L-NAME rats before and after ischemia (*n* = 6). Data were analysed using one-way ANOVA followed by Bonferroni’s post hoc test. ^Ψ^
*p* < 0.05 vs. pre-ischemic phase of respective groups; * *p* < 0.05 vs. WKY-CONTROL at pre-ischemic phase; ^φ^
*p* < 0.05 vs. WKY-CONTROL at reperfusion phase; ^δ^
*p*< 0.05 vs. WKY+NaHS at reperfusion phase; ^#^
*p* < 0.05 vs. L-NAME-CONTROL at reperfusion phase; ^ᴫ^
*p* < 0.05 vs. L-NAME+NaHS at reperfusion phase. MDA = malondialdehyde, T-SOD = total superoxide dismutase, GSH = glutathione, PAG = dL-propargylglycine, WKY = *Wistar Kyoto* rats, L-NAME = L-nitro-methyl-arginine-ester.

**Figure 5 biomolecules-11-01549-f005:**
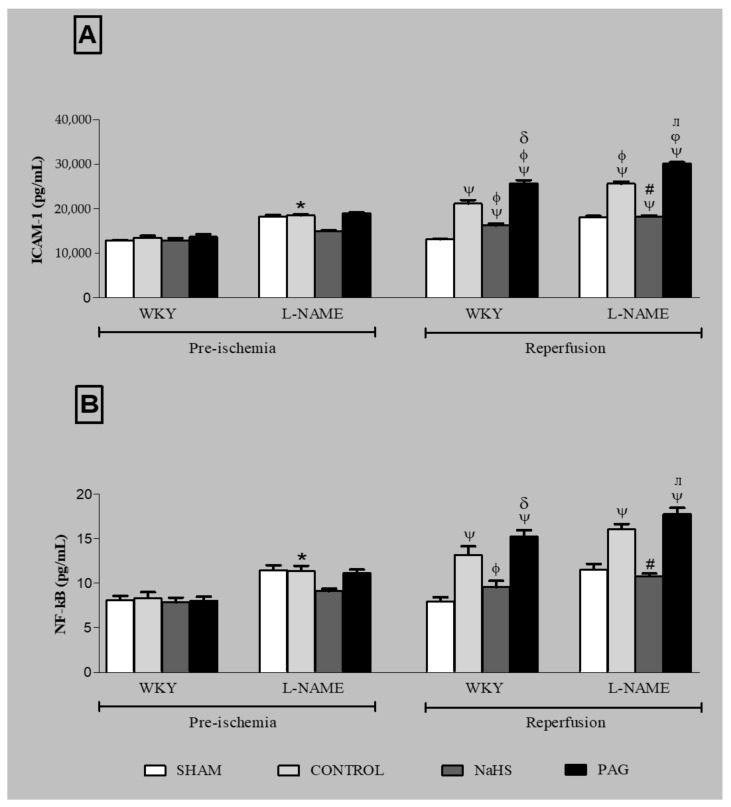
ICAM-1 concentration (**A**) and NF-kB concentration (**B**) in the kidney tissues of different experimental groups of WKY and L-NAME rats before and after ischemia (*n* = 6). Data were analysed using one-way ANOVA followed by Bonferroni’s post hoc test. ^Ψ^
*p* < 0.05 vs. pre-ischemic phase of respective groups; * *p* < 0.05 vs. WKY-CONTROL at pre-ischemic phase; ^φ^
*p* < 0.05 vs. WKY-CONTROL at reperfusion phase; ^δ^
*p*< 0.05 vs. WKY+NaHS at reperfusion phase; ^#^
*p* < 0.05 vs. L-NAME-CONTROL at reperfusion phase; ^ᴫ^
*p* < 0.05 vs. L-NAME+NaHS at reperfusion phase. ICAM-1 = intercellular adhesion molecule-1. NF-kB = nuclear factor-kappa B, PAG = dL-propargylglycine, WKY = *Wistar Kyoto* rats, L-NAME = L-nitro-methyl-arginine-ester.

**Figure 6 biomolecules-11-01549-f006:**
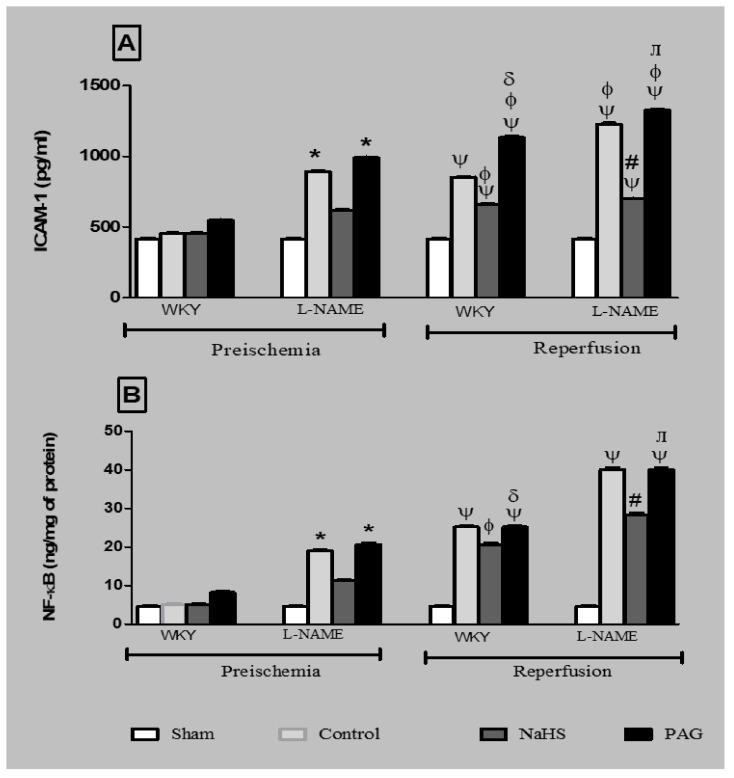
(**A**) ICAM-1 concentration and (**B**) NF-kB concentration in kidney tissues of different experimental groups of WKY and L-NAME rats before and after ischemia (*n* = 6). Data were analysed using one-way ANOVA followed by Bonferroni’s post hoc test. ^Ψ^
*p* < 0.05 vs. pre-ischemic phase of respective groups; * *p* < 0.05 vs. WKY-CONTROL at pre-ischemic phase; ^φ^
*p* < 0.05 vs. WKY-CONTROL at reperfusion phase; ^δ^
*p*< 0.05 vs. WKY+NaHS at reperfusion phase; ^#^
*p* < 0.05 vs. L-NAME-CONTROL at reperfusion phase; ^ᴫ^
*p* < 0.05 vs. L-NAME+NaHS at reperfusion phase. ICAM-1 = intercellular adhesion molecule-1; NF-kB = nuclear factor-kappa B; PAG = dL-propargylglycine; WKY = *Wistar Kyoto* rats; L-NAME = L-nitro-methyl-arginine-ester.

**Figure 7 biomolecules-11-01549-f007:**
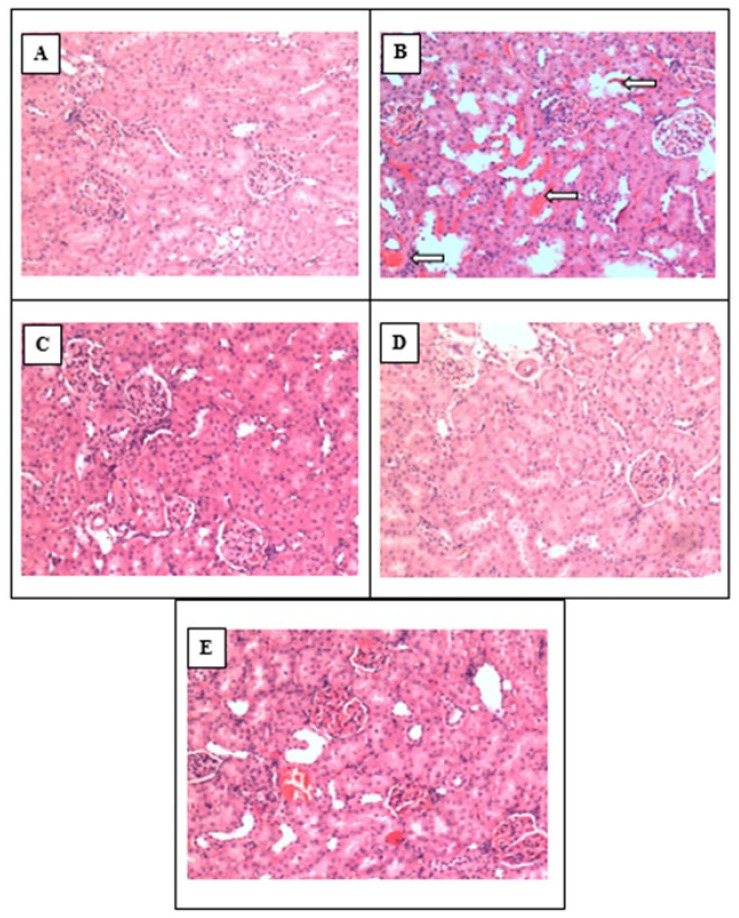
Histopathology study of kidneys from (**A**) control, (**B**) renal ischemic-reperfusion injury, (**C**) sham-control, (**D**) H_2_S-treated and (**E**) renal ischemic-reperfusion injury treated with H_2_S rats (magnification: 10×).

**Table 1 biomolecules-11-01549-t001:** Hemodynamics and renal functional parameters in all experimental groups of WKY and L-NAME rats during metabolic collection on days 0, 21 and 35.

Parameters	Groups	Day 0	Day 21	Day 35
**SBP**	WKY-SHAM	117 ± 3	118 ± 3	119 ± 3
	WKY-CONTROL	117 ± 5	120 ± 3	119 ± 2
	WKY+NaHS	121 ± 3	117 ± 3	113 ± 2
	WKY+PAG	119 ± 4	116 ± 3	117 ± 3
	L-NAME-SHAM	118 ± 4	162 ± 4 ^Ψ^	199 ± 9 ^Ψ,φ^
	L-NAME-CONTROL	119 ± 3	180 ± 4 ^Ψ^	194 ± 9 ^Ψ,φ,δ^
	L-NAME+NaHS	118 ± 2	139 ± 3 ^Ψ^	154 ± 3 ^Ψ,φ,ᴫ^
	L-NAME+PAG	116 ± 3	173 ± 6 ^Ψ^	197 ± 10 ^Ψ,φ^
**MAP**	WKY-SHAM	98 ± 2	99 ± 2	100 ± 3
	WKY-CONTROL	99 ± 4	98 ± 1	99 ± 1
	WKY+NaHS	100 ± 2	99 ± 2	95 ± 1
	WKY+PAG	100 ± 2	98 ± 1	98 ± 2
	L-NAME-SHAM	99 ± 2	142 ± 3 ^Ψ^	176 ± 6 ^Ψ,φ^
	L-NAME-CONTROL	99 ± 2	144 ± 3 ^Ψ^	177 ± 10 ^Ψ,φ,δ^
	L-NAME+NaHS	99 ± 1	120 ± 2 ^Ψ^	141 ± 8 ^Ψ,φ,ᴫ^
	L-NAME+PAG	98 ± 1	142 ± 4 ^Ψ^	175 ± 6 ^Ψ,φ^
**HR**	WKY-SHAM	326 ± 12	311 ± 9	319 ± 8
	WKY-CONTROL	326 ± 15	331 ± 12	324 ± 11
	WKY+NaHS	313 ± 14	304 ± 9	321 ± 7
	WKY+PAG	308 ± 19	304 ± 20	318 ± 12
	L-NAME-SHAM	303 ± 21	338 ± 32	384 ± 15 ^Ψ,φ^
	L-NAME-CONTROL	308 ± 12	340 ± 16	387 ± 18 ^Ψ,φ,δ^
	L-NAME+NaHS	310 ± 7	338 ± 7	355 ± 13 ^Ψ^
	L-NAME+PAG	300 ± 12	327 ± 10	394 ± 15 ^Ψ,φ^
**FE_Na_**	WKY-SHAM WKY-CONTROL	0.72 ± 0.09 0.67 ± 0.04	0.69 ± 0.1 0.66 ± 0.1	0.74 ± 0.07 0.69 ± 0.16
	WKY+NaHS	0.67 ± 0.02	0.68 ± 0.02	0.69 ± 0.01
	WKY+PAG	0.65 ± 0.02	0.68 ± 0.02	0.68 ± 0.03
	L-NAME-SHAM	0.72 ± 0.02	0.96 ± 0.03 ^Ψ^	1.51 ± 0.06 ^Ψ,φ^
	L-NAME-CONTROL	0.68 ± 0.03	0.98 ± 0.03 ^Ψ^	1.56 ± 0.05 ^Ψ,φ,δ^
	L-NAME+NaHS	0.72 ± 0.02	0.84 ± 0.02	0.94 ± 0.02 ^Ψ,ᴫ^
	L-NAME+PAG	0.69 ± 0.03	0.96 ± 0.03 ^Ψ^	1.61 ± 0.08 ^Ψ,φ^
**FE_K_**	WKY-SHAM	14 ± 3.2	13 ± 2.9	13 ± 2.0
	WKY-CONTROL	13 ± 1.1	13 ± 5.0	13 ± 3.0
	WKY+NaHS	13 ± 2.8	12.09 ± 1.3	12 ± 1.6
	WKY+PAG	12 ± 1.8	12 ± 1.3	11 ± 1.1
	L-NAME-SHAM	13 ± 1.9	18 ± 3.0 ^Ψ^	27 ± 3.9 ^Ψ,φ^
	L-NAME-CONTROL	12 ± 1.8	18 ± 2.9 ^Ψ^	27 ± 4.1 ^Ψ,φ,δ^
	L-NAME+NaHS	13 ± 1.9	15 ± 2.2	19 ± 2.3 ^Ψ,ᴫ^
	L-NAME+PAG	12 ± 1.9	16 ± 1.4 ^Ψ^	25 ± 3.6 ^Ψ,φ^
**Cr.Cl**	WKY-SHAM	5.36 ± 0.62	5.21 ± 1.02	4.63 ± 0.54
	WKY-CONTROL	7.82 ± 1.42	7.12 ± 1.59	5.64 ± 0.58
	WKY+NaHS	8.06 ± 0.41	6.51 ± 2.42	6.71 ± 1.52
	WKY+PAG	4.70 ± 0.68	4.06 ± 0.55	5.67 ± 0.80
	L-NAME-SHAM	5.24 ± 0.3	2.53 ± 0.3 ^Ψ^	1.61 ± 0.2 ^Ψ^
	L-NAME-CONTROL	7.87 ± 3.0	4.49 ± 2.1 ^Ψ^	3.09 ± 1.3 ^Ψ,δ^
	L-NAME+NaHS	4.83 ± 0.5	3.41 ± 0.2	2.93 ± 0.3
	L-NAME+PAG	5.63 ± 0.9	3.45 ± 1.0 ^Ψ^	2.56 ± 0.6 ^Ψ^

Data were analysed using repeated measures ANOVA followed by Bonferroni’s post hoc test. Data are presented as means ± SEM (*n* = 6, per group). ^Ψ^
*p* < 0.05 vs. day 0 of respective groups; ^φ^
*p* < 0.05 vs. day 21 of respective groups; ^δ^
*p* < 0.05 vs. WKY-CONTROL on day 35; ^ᴫ^
*p* < 0.05 vs. L-NAME-CONTROL on day 35. SBP = systolic blood pressure, MAP = mean arterial pressure, HR = heart rate. WKY = *Wistar Kyoto* rats, L-NAME = L-nitro-methyl- arginine-ester, PAG = dL-propargylglycine.

**Table 2 biomolecules-11-01549-t002:** Hemodynamics, kidney weight and kidney index for all experimental groups of WKY and L-NAME rats on acute experiment day (day 36).

Groups	SBP (mmHg)	MAP (mmHg)	HR (rpm)	BW (g)	KW (g)	KI (%)
WKY-SHAM	113 ± 5	100 ± 4	316 ± 9	305 ± 10	0.90 ± 0.01	0.297 ± 0.01
WKY-CONTROL	111 ± 6	99 ± 2	311 ± 14	315 ± 10	1.06 ± 0.04 ^Ψ^	0.336 ± 0.02 ^Ψ^
WKY+NaHS	110 ± 4	98 ± 3	300 ± 8	298 ± 7 ^#^	0.94 ± 0.01 ^#^	0.317 ± 0.00
WKY+PAG	113 ± 3	99 ± 1	318 ± 20	302 ± 5	1.03 ± 0.05 ^Ψ,δ^	0.340 ± 0.02 ^Ψ^
L-NAME-SHAM	198 ± 9	175 ± 6	374 ± 16	294 ± 6	0.98 ± 0.02	0.332 ± 0.01
L-NAME-CONTROL	193± 9 ^#^	176 ± 10 ^#^	379 ± 23 ^#^	285 ± 7 ^#^	1.09 ± 0.05 ^φ^	0.384 ± 0.02 ^φ,#^
L-NAME+NaHS	153 ± 3 ^φ,¥^	140 ± 8 ^φ,¥^	352 ± 18	297 ± 4	1.04 ± 0.06	0.349 ± 0.01
L-NAME+PAG	196 ± 10 ^ᴫ^	174 ± 6 ^ᴫ^	389 ± 14 ^ᴫ^	285 ± 7	1.174 ± 0.02 ^φ,¥,ᴫ^	0.412 ± 0.01 ^φ,ᴫ^

Data were analysed using one-way ANOVA followed by Bonferroni’s post hoc test. Data presented as means ± SEM (*n* = 6, per group). ^Ψ^
*p* < 0.05 vs. WKY-SHAM; ^φ^
*p* < 0.05 vs. L-NAME-SHAM; ^#^
*p* < 0.05 vs. WKY-CONTROL; ^δ^
*p* < 0.05 vs. WKY+NaHS; ^¥^
*p* < 0.05 vs. L-NAME-CONTROL; ^ᴫ^
*p* < 0.05 vs. L-NAME+NaHS. SBP = systolic blood pressure, MAP = mean arterial pressure, HR = heart rate, BW = body weight, KW = kidney weight, KI = kidney index, WKY = Wistar Kyoto rats, L-NAME = L-nitro-methyl-arginine-ester, PAG = dL-propargylglycine.

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
