# Peer review of "Hydrogen Sulphide Treatment Prevents Renal Ischemia-Reperfusion Injury by Inhibiting the Expression of ICAM-1 and NF-kB Concentration in Normotensive and Hypertensive Rats"

_biomolecules, 2021, doi:10.3390/biom11101549_

Round 1

Reviewer 1 Report

The authors have addressed my concerns regarding renal parenchymal expression of ICAM1 and NFkB and renal histology.

Fig. 7. I am not convinced there is glomerular hypertrophy and the authors do not provide morphometric data. They should remove any reference to glomerular hypertrophy. 

P value is wrongly shown on line 445.

Author Response

Reviewer 1:

Comment 1 of the Reviewer 1:

Fig. 7. I am not convinced there is glomerular hypertrophy and the authors do not provide morphometric data. They should remove any reference to glomerular hypertrophy. 

Response by author:

All the changes have been done as suggested reviewer 1. Glomerular hypertrophy is removed in the result section of the histopathology while the black arrows showing Glomerular hypertrophy in Fig. 7-B have also been removed.

Comment 2 of the Reviewer 1:

P value is wrongly shown on line 445.

Response by author:

P value is corrected on line number 445 as suggested by reviewer 1.

Reviewer 2 Report

The changings made by the authors improved the quality of the manuscript. However, some issues remained. In particular, the results section is a bit confusing, especially in the first paragraphs (i.e. Systemic hemodynamic parameters, Renal functional parameters). For example, the sentence “In WKY groups no significant change (all P> 0.05) in SBP, MAP and HR was observed on day 21 when compared to their respective groups on day 0 as well as on day 35 when compared to their respective groups on both days 0 and 21.” is not clear. The authors should work on the “results” section in order to make it more clear in fluent. Conclusions need to be extended too.

Minor corrections:

-“grams” abbreviation is “g” not “gm”. Please, correct throughout the manuscript;

-line 208: please correct kidney “index” and the formula ( “x 100”)

Author Response

Reviewer 2:

The changings made by the authors improved the quality of the manuscript. However, some issues remained. In particular, the results section is a bit confusing, especially in the first paragraphs (i.e. Systemic hemodynamic parameters, Renal functional parameters). For example, the sentence “In WKY groups no significant change (all P> 0.05) in SBP, MAP and HR was observed on day 21 when compared to their respective groups on day 0 as well as on day 35 when compared to their respective groups on both days 0 and 21.” is not clear. The authors should work on the “results” section in order to make it more clear in fluent. Conclusions need to be extended too.

Response by author:

Author appreciate the comments of reviewer 2 by highlighting ambiguous presentation of the result. Mentioned paragraph is rephrased and updated version of the manuscript is made it easy to understand. Conclusion part is also extended with on the basis of current study findings.

This manuscript is a resubmission of an earlier submission. The following is a list of the peer review reports and author responses from that submission.

Round 1

Reviewer 1 Report

Authors aim to study the mechanism of amelioration of kidney ischemia-reperfusion injury (IRI) in normotensive and hypertensive rats treated with hydrogen sulfide (H2S). The following major concerns need attention.

  1. Methods: Previous publications (Bos 2009, 2013) have shown that administration of NaHS minutes before IRI was enough to mitigate injury. What is the rationale for weeks of NaHS administration?
  2. Methods: Is it okay to clamp only one kidney to create IRI? Note the animals were euthanized at the end of 3 hours of reperfusion following 30 min ischemia.
  3. Methods: What is the basis for measuring NFkB, ICAM-1, and oxidative parameters in the plasma and not the kidney, where the injury is occurring? ICAM-1 is generally expressed by the endothelial cells. NFkB is activated in kidney cells during injury. IRI causes oxidative damage to the kidney. Therefore, changes in these parameters in the kidney would be of relevance.
  4. Table 1. Dietary Na and K intake affects their urinary excretion. Did the authors measure dietary Na and K intake in the rats among the various groups? How do they know that the rats in different groups ingested the same amount of Na and K so that one can compare their FENa and FEK data?
  5. Table 1. Creatinine clearance fell by nearly 70% in L-NAME sham rats on day 35. This means serum creatinine more than doubled in this group. However, this was not the case in L-NAME Control group. Why?
  6. 3. It is striking that no change in serum creatinine occurred following ischemia and reperfusion in either WKY or L-NAME induced hypertensive rats relative to their sham operated controls.
  7. Authors do not provide any histologic evidence of ischemic necrosis and markers of inflammation in the kidneys.
  8. The manuscript should be edited thoroughly for errors in grammar.

Author Response

Author acknowledgement:

Corresponding author on the behalf of all co-authors pay gratitude to the editor for generously considering this manuscript for review and authors also thank the reviewer 1 for his positive comments to improve the quality of this manuscript to attract the reader’s community of biomolecules journal of the MDPI publisher. We have again put our energy to meet the expectations of the editor and reviewer for the publication of this manuscript. However, author welcomes any further suggestion to improve the quality of manuscript.

Reviewer 1.

Authors aim to study the mechanism of amelioration of kidney ischemia-reperfusion injury (IRI) in normotensive and hypertensive rats treated with hydrogen sulfide (H2S). The following major concerns need attention.

Comment 1.

Methods: Previous publications (Bos 2009, 2013) have shown that administration of NaHS minutes before IRI was enough to mitigate injury. What is the rationale for weeks of NaHS administration?

Response to comment

Author appreciate the comments of reviewer 1 by pointing the previously reported data to clear objectives of present study. Previously mentioned study is also cited in updated version of current manuscript which indeed will complement the literature reported previously on the role of H2S in ischemia reperfusion injury. Secondly, reported study used higher doses of H2S 100 ppm-500 ppm which are toxic doses and start producing hypoxia. Authors of previously reported study mentioned that this hypoxia produced by H2S prior to I/R is good enough to provide protection against IRI. Present study, used physiological doses of H2S (56 µmol/kg) in chronic study by using same concept of supplementation of H2S prior to I/R, we maintained the Physiological level of the H2S and explored that prophylactic uses of H2S donor prior to I/R can protect the kidney damage in organ transplant procedure.

Second reason for chronic administration of H2S is augmentation of antioxidant status of the body and bringing oxidative stress down so that injury during I/R should not be able to dominate oxidative stress over antioxidant status of the body by suppressing the NF-KB and ICAM. Supplementation of H2S few minutes prior to IRI may not provide as good antioxidant status to the body as with chronic study.

Comment 2.

Methods: Is it okay to clamp only one kidney to create IRI? Note the animals were euthanized at the end of 3 hours of reperfusion following 30 min ischemia.

Response to comment

Authors appreciate the brain storming question on one kidney clamoed method for the creation of IRI. In fact, one kidney was clamped to make animal model closer to clinical setting of kidney transplant where mostly one kidney is transplanted while other kidney is intact. So we concluded that in one kidney transplant, IRI can be minimize by chronic supplementation of H2S in normotensive and hypertensive patients while second kidney was kept away from all interventions. 

Comment 3.

Methods: What is the basis for measuring NF-kB, ICAM-1, and oxidative parameters in the plasma and not the kidney, where the injury is occurring? ICAM-1 is generally expressed by the endothelial cells. NF-kB is activated in kidney cells during injury. IRI causes oxidative damage to the kidney. Therefore, changes in these parameters in the kidney would be of relevance.

Response to comment

Author again convinced with logical question raised by the reviewer 1. It would be pin point conclusion if we measure NF-KB and ICAM-1 in the kidney undergoing ischemia reperfusion but again question would be raised about the status of 2nd kidney. We measured global or systemic concentrations of NF-KB and ICAM-1 which not only gives us regional picture but also giving us insight that regional IRI in one kidney raised the NF-KB and ICAM-1 globally which might affect the second kidney. However, this comment of reviewer 1 give us directions for future studies that we should measure regional as well as global levels of these markers which will give us accumulated informations.

Comment 4

Table 1. Dietary Na and K intake affects their urinary excretion. Did the authors measure dietary Na and K intake in the rats among the various groups? How do they know that the rats in different groups ingested the same amount of Na and K so that one can compare their FENa and FEK data?

Response to comment

Dietry intake was kept standard by adding standard rodent chow (Gold Coin Sdn. Bhd., Penang, Malaysia) to all animals containing 3% fat, 22% protein and 5% fibers. This food was added in our previously data to report the effect of fructose food diet on kidney (1).

  1. Mohammed H. Abdulla • Munavvar A. Sattar • Nor A. Abdullah • Md. Abdul Hye Khan • Kolla R. L. Anand Swarup • Edward J. Johns. The contribution of a1B-adrenoceptor subtype in the renal vasculature of fructose-fed Sprague–Dawley rats. Eur J Nutr (2011) 50:251–260).

Comment 5

Table 1. Creatinine clearance fell by nearly 70% in L-NAME sham rats on day 35. This means serum creatinine more than doubled in this group. However, this was not the case in L-NAME Control group. Why?

Response to comment

Creatinine clearance was dropped by 70 % in L-NAME sham and 50 % L-NAME Control group rats on day 35 when compared to day 0 which clearly indicate that NO blockade might have caused preglomerulus vasoconstriction leading to decrease creatinine clearance which further can be seen that NO blockade in same groups has reduced renal cortical blood pressure on day 35 as shown in Fig 2. It indicate that blockade of NO in L-NAME sham and L-NAME control reduced the Cr.Cl and RCBP when compared to WKY-Sham and Wky- Control groups.

Comment 6

It is striking that no change in serum creatinine occurred following ischemia and reperfusion in either WKY or L-NAME induced hypertensive rats relative to their sham operated controls.

Response to comment

Chronic administration of L-NAME for 35 days resulted in significant difference in serum creatinine levels in WKY and L-NAME groups but no major difference was observed in pre-ischemic and reperfusion phase. This might explain the short duration of acute studies which may not be good enough to see significant changes. Only significant changes in ischemia and reperfusion phases were observed change in RCBP, H2S concentration, ICAM-1 and NF-KB in ischemia and reperfusion phases which may lead to the compromised kidney function.

Comment 7

Authors do not provide any histologic evidence of ischemic necrosis and markers of inflammation in the kidneys.

Response to comment

Author appreciate the reviewer 1 comments for raising the question about lack of histopathology slides which is short coming of present study. Despite of having all the tissues we were unable to produce histopathological slides due to tissue transportation to histopathological slides facility area due to Covid-19, restriction in working condition in laboratory in Malaysia, complete closure of the country and off-campus education. All these factors still persists and we don’t want to hold our full scientific data for histopathological slides though very important for study.

Comment 8

The manuscript should be edited thoroughly for errors in grammar.

Response to comment

Whole manuscript is read by Professor Edward James John who is eminent scientist of kidney domain and native speaker of English language and affiliated in University College cork.

Reviewer 2 Report

The present work aims to investigate the effect of chronic administration of sodium hydrosulphide as H2S donor in the attenuation of renal ischemia reperfusion injury in normotensive and L-NAME induced hypertensive rats. The context and the rational of the work were clearly described, as well as the methodologies used, meticulously described by the authors. Although the subject has been object of intense study from many research groups, with many papers published on this topic, the authors highlighted the elements of novelties. The results obtained were critically reported, as well as the potential applications in clinics. However, when discussing the anti-oxidants and anti-inflammatory effects exerted by exogenous administration of H2S, the authors should also mention the possible contribution of endogenous Carbon Monoxide production to the observed renoprotective effects involving the NF-kB pathway, and the interplay and crosstalk between H2S, NO and CO. See "Aziz NM, Elbassuoni EA, Kamel MY, Ahmed SM. Hydrogen sulfide renal protective effects: possible link between hydrogen sulfide and endogenous carbon monoxide in a rat model of renal injury. Cell Stress Chaperones. 2020 Mar;25(2):211-221. doi: 10.1007/s12192-019-01055-2."  Conclusions should be also extended. Minor corrections are also needed before the paper can be accepted for publication:

Line 87: the unit of animal weight is missing. Please correct.

Line 177: equation should be corrected (? 00)

Line 191: please correct “N,N-Dimethyl-p-phenylenediamine sulfate”

Line 311: the sentence is pending. Please, continue.

The quality of figures 1-5 should be definitely improved.

More abbreviations are needed in the abbreviation list.

Please, check the reference style : see Refs 36 and 41.

Author Response

Author acknowledgement:

Corresponding author on the behalf of all co-authors pay gratitude to the editor for generously considering this manuscript for review and authors also thank the reviewer 2 for his positive comments to improve the quality of this manuscript to attract the reader’s community of biomolecules journal of the MDPI publisher. We have again put our energy to meet the expectations of the editor and reviewer for the publication of this manuscript. However, author welcomes any further suggestion to improve the quality of manuscript.

Comment 1.

The present work aims to investigate the effect of chronic administration of sodium hydrosulphide as H2S donor in the attenuation of renal ischemia reperfusion injury in normotensive and L-NAME induced hypertensive rats. The context and the rational of the work were clearly described, as well as the methodologies used, meticulously described by the authors. Although the subject has been object of intense study from many research groups, with many papers published on this topic, the authors highlighted the elements of novelties. The results obtained were critically reported, as well as the potential applications in clinics. However, when discussing the anti-oxidants and anti-inflammatory effects exerted by exogenous administration of H2S, the authors should also mention the possible contribution of endogenous Carbon Monoxide production to the observed renoprotective effects involving the NF-kB pathway, and the interplay and crosstalk between H2S, NO and CO. See "Aziz NM, Elbassuoni EA, Kamel MY, Ahmed SM. Hydrogen sulfide renal protective effects: possible link between hydrogen sulfide and endogenous carbon monoxide in a rat model of renal injury. Cell Stress Chaperones. 2020 Mar; 25(2):211-221. doi: 10.1007/s12192-019-01055-2."  Conclusions should be also extended. Minor corrections are also needed before the paper can be accepted for publication:

Response to comment

Authors appreciate for the suggestions of reviewer 2 and helped us to include one elegant study in bibliography which link H2S and CO. This study is included as reference number 31.

Comment 2.

Line 87: the unit of animal weight is missing. Please correct.

Response to comment

Corrected as suggested by the reviewer 2.

Comment 3.

Line 177: equation should be corrected (? 00)

Response to comment

Corrected as suggested by the reviewer 2.

Comment 4.

Line 191: please correct “N,N-Dimethyl-p-phenylenediamine sulfate”

Response to comment

Corrected as suggested by the reviewer 2.

Comment 5.

Line 311: the sentence is pending. Please, continue.

Response to comment

Following sentence was added as suggested by reviewer 2 and we thank him for highlighting a major point.’’

‘’No significant effect (P> 0.05) of NaHS was observed on RCBP in WKY+NaHS group when compared to WKY-CONTROL and WKY+PAG groups. But on the other hand, significant increase (P< 0.05) in RCBP was found in L-NAME+NaHS group when compared to WKY-CONTROL and WKY+PAG groups. It was noticed that RCBP was significantly lower (P< 0.05) in L-NAME induced hypertensive rats when compared to WKY rats.’’

Comment 6.

The quality of figures 1-5 should be definitely improved.

Response to comment

Quality of the all the Fig. 1-5 are improved in the updated version of manuscript as suggested by Reviewer 2.

Comment 7.

More abbreviations are needed in the abbreviation list.

Response to comment

As suggested by reviewer 2, 19 more abbreviation have been added in updated version of manuscript.

Comment 8.

Please, check the reference style: see Refs 36 and 41.

Response to comment

References 36 which is 38 and 41 which is 43 in updated version of manuscript has been corrected as suggested by the reviewer 2.

Round 2

Reviewer 1 Report

The authors have endeavored to respond to concerns raised in the first review by this reviewer. However, many of the concerns remain.

Response to Comment 1: The authors state that Bos et al employed ‘toxic’ doses of H2S. However, that dose of H2S was well tolerated by the mice and IRI improved. If the authors wanted to ensure antioxidant actions of H2S was achieved, they should have done a time course to find the minimum duration of H2S administration required to achieve this response.

Response to Comment 3: The authors’ reply is not convincing. Measurement of systemic parameters does not give an accurate picture of events in the kidney.

Response to Comment 4: It appears that the authors do not have directly measured data on food consumption and dietary Na and K intake that can be compared between the groups.

Response to Comment 7: While I appreciate the difficulty posed by COVID in conducting experiments, still the question remains whether the protocol employed by the authors resulted in significant ischemic injury in the kidney.